# Inhibitors of Mitochondrial Human Carbonic Anhydrases VA and VB as a Therapeutic Strategy against Paclitaxel-Induced Neuropathic Pain in Mice

**DOI:** 10.3390/ijms23116229

**Published:** 2022-06-02

**Authors:** Laura Micheli, Lara Testai, Andrea Angeli, Donatello Carrino, Alessandra Pacini, Francesco Margiotta, Lorenzo Flori, Claudiu T. Supuran, Vincenzo Calderone, Carla Ghelardini, Lorenzo Di Cesare Mannelli

**Affiliations:** 1Department of Neuroscience, Psychology, Drug Research and Child Health, NEUROFARBA-Pharmacology and Toxicology Section, University of Florence, Viale Pieraccini 6, 50139 Florence, Italy; francesco.margiotta@unifi.it (F.M.); carla.ghelardini@unifi.it (C.G.); lorenzo.mannelli@unifi.it (L.D.C.M.); 2Department of Pharmacy, University of Pisa, Via Bonanno 6, 56126 Pisa, Italy; lara.testai@unipi.it (L.T.); lorenzo.flori@unipi.it (L.F.); vincenzo.calderone@unipi.it (V.C.); 3Department of Neuroscience, Psychology, Drug Research and Child Health, Pharmaceutical and Nutraceutical Sciences Section, University of Florence, Via Ugo Schiff 6, 50019 Florence, Italy; andrea.angeli@unifi.it (A.A.); claudiu.supuran@unifi.it (C.T.S.); 4Department of Experimental and Clinical Medicine, Anatomy and Histology Section, University of Florence, Largo Brambilla 3, 50134 Florence, Italy; donatello.carrino@unifi.it (D.C.); alessandra.pacini@unifi.it (A.P.)

**Keywords:** neuropathic pain, *Taxus brevifolia*, carbonic anhydrase, CA VA and VB, mitochondria, glial cells, neuroprotection

## Abstract

Neuropathy development is a major dose-limiting side effect of anticancer treatments that significantly reduces patient’s quality of life. The inadequate pharmacological approaches for neuropathic pain management warrant the identification of novel therapeutic targets. Mitochondrial dysfunctions that lead to reactive oxygen species (ROS) increase, cytosolic Ca^2+^ imbalance, and lactate acidosis are implicated in neuropathic pain pathogenesis. It has been observed that in these deregulations, a pivotal role is played by the mitochondrial carbonic anhydrases (CA) VA and VB isoforms. Hence, preclinical studies should be conducted to assess the efficacy of two novel selenides bearing benzenesulfonamide moieties, named **5b** and **5d**, and able to inhibit CA VA and VB against paclitaxel-induced neurotoxicity in mice. Acute treatment with **5b** and **5d** (30–100 mg/kg, per os – p.o.) determined a dose-dependent and long-lasting anti-hyperalgesic effect in the Cold plate test. Further, repeated daily treatment for 15 days with 100 mg/kg of both compounds (starting the first day of paclitaxel injection) significantly prevented neuropathic pain development without the onset of tolerance to the anti-hyperalgesic effect. In both experiments, acetazolamide (AAZ, 100 mg/kg, p.o.) used as the reference drug was partially active. Moreover, ex vivo analysis demonstrated the efficacy of **5b** and **5d** repeated treatments in reducing the maladaptive plasticity that occurs to glia cells in the lumbar portion of the spinal cord and in improving mitochondrial functions in the brain and spinal cord that were strongly impaired by paclitaxel-repeated treatment. In this regard, **5b** and **5d** ameliorated the metabolic activity, as observed by the increase in citrate synthase activity, and preserved an optimal mitochondrial membrane potential (ΔΨ) value, which appeared depolarized in brains from paclitaxel-treated animals. In conclusion, **5b** and **5d** have therapeutic and protective effects against paclitaxel-induced neuropathy without tolerance development. Moreover, **5b** and **5d** reduced glial cell activation and mitochondrial dysfunction in the central nervous system, being a promising candidate for the management of neuropathic pain and neurotoxicity evoked by chemotherapeutic drugs.

## 1. Introduction

Paclitaxel is a chemotherapeutic agent derived from the bark of *Taxus brevifolia* and is frequently used against a broad spectrum of tumors [1].

Unfortunately, according to the dose intensity [2,3], paclitaxel frequently induces neuropathic pain that severely impair the patient’s quality of life, leading to treatment discontinuation [4]. Most frequently, the clinical picture is dominated by severe symptoms such as tingling, numbness, spontaneous pain, and referred pain to mechanical and thermal stimuli in the hands and feet of treated patients [5,6].

Neuropathic pain treatment has two distinct targets, the prevention of the development of the disease and the management of the established pathology. To date, the effective management of neuropathic pain is a clinical need still unmet and challenging since existing approaches are far from suitable due to their limited effectiveness and adverse side effects profiles [7].

Recent discoveries, aimed to identify and characterize new therapeutic strategies, have highlighted the inhibition of carbonic anhydrase (CA) as a new valid approach for pain management [8]. Carbonic anhydrase inhibitors (CAIs) have been shown to possess pain-relieving properties against inflammatory pain resembling human rheumatoid arthritis [9,10,11] against persistent visceral pain, in which the CA IV isoform is particularly involved, [12] as well as against oxaliplatin-induced pain threshold alterations [13,14]. CAs are a ubiquitous superfamily of enzymes that catalyze a reaction fundamental for life: the hydration of CO_2_ to bicarbonate and protons. These enzymes regulate a wide range of physiological processes in a variety of tissues and cell compartments while an overexpression of CA is frequently linked to pathological illnesses such as glaucoma [15], obesity [16], tumorigenesis [17,18,19], and epilepsy [20].

Among the eight unique CA families known, the α-family is the most widely studied and comprises 16 members, each characterized for tissue-specific expression and cellular and sub-cellular localization. Under neuropathic pain conditions, the mitochondrial CA VA and CA VB isoforms are of particular interest. It has been shown that mitochondrial CA VA/VB regulate the respiration rate as well as reactive oxygen species (ROS) production, oxidative stress, and apoptosis [21], and mitochondrial dysfunction is implicated in chemotherapy-induced neuropathic pain [22]. Indeed, mitochondria are involved in many essential functions, including ATP production through oxidative phosphorylation, apoptosis regulation, intracellular calcium homeostasis, and ROS production. Given their pivotal role, any alteration to mitochondrial integrity and functionality can impact cellular functionality, leading to disorders. Many factors are involved in the maintenance of mitochondrial activity, and during the last decade, a large body of evidence indicates that chemotherapeutic drugs determine mitochondrial injury characterized by loss of mitochondrial morphology and disruption of oxidative phosphorylation and mitochondrial membrane potential. All of these events determine a reduction in ATP production and increase of the reliance on glycolysis, causing a decrease in the cellular bio-energetic capacity [22]. Moreover, an uncontrolled release of ROS and reactive nitrogen species (RNS) evoked by antioxidant enzyme deregulation may cause oxidative stress and nitrosylative and nitrative reactions with proteins and nucleic acids, and these phenomena can be strongly involved in the onset of neuropathic pain induced by chemotherapy [23,24,25,26]. Furthermore, very recently, altered levels of mitochondrial DNA and complex I activity have been proposed as potential blood biomarkers for chemotherapy-induced neuropathic pain [27]. The mitochondrial CA V isoform is expressed in the nervous tissue and in particular in astrocytes and neurons, suggesting a cell-specific, physiological role. In astrocytes, CA V acts as an important player in gluconeogenesis while in neurons is involved in the regulation of the intramitochondrial calcium level, contributing to the stability of the cells. Moreover, CA V also regulates the bicarbonate homeostasis in neurons, contributing to explain some neurotrophic effects of CAIs [28]. In addition, CA V regulates the rate of ROS production, and its inhibition has been shown to rescue the mouse brain from glucose-induced pericyte loss [29]. Therefore, although other CA isoforms have been proved to be closely involved in neuropathic pain management, in this work, we explored the efficacy of two selected CAIs particularly active against the CA VA and VB isoforms, named **5b** and **5d** and reported in Figure 1, in a mouse model of paclitaxel-induced neuropathic pain. In particular, the anti-hypersensitivity effects of acute and sub-chronic treatment were evaluated; moreover, ex vivo analysis of the central nervous system was assessed to determine the effects of the two selected molecules on the mitochondrial membrane potential (ΔΨ) value and on the citrate synthase activity. Moreover, the glial cell profile in the dorsal horn of the lumbar spinal cord was determined.

## 2. Results and Discussion

Presently, no valid pharmacological strategies are available once patients develop chemotherapy-induced neuropathy. The symptomatic approach recommends the use of off-label drugs such as amitryptyline [30] with, unfortunately, not satisfying effectiveness [31]. In this context, the possibility to effectively manage or prevent neurotoxicity by the oral administration of “rescue drugs” could have a great clinical impact leading to significantly increase the patient’ quality of life that usually is strongly affected.

In recent years, carbonic anhydrase (CA) has gained growing attention since CA isoforms are displaced in many tissues and organs and are involved in different physiological and pathological processes [32].

Previous works have highlighted the active role of several carbonic anhydrase inhibitors (CAIs) in reducing different pain states in preclinical settings. CAIs’ efficacy against rheumatoid arthritis has emerged [9,10,11] as well as against visceral pain, and in this experiment, the CAI IV isoform has been revealed as the main actor to counteract the development of colitis in rats [12]. Moving to neuropathies induced by chemotherapeutic drugs, CAIs have been observed to possess anti-neuropathic properties in particular against oxaliplatin-induced neurotoxicity [13,14,33]. Nevertheless, no information is available regarding the pain relieving properties of CAIs against paclitaxel-induced neurotoxicity. In the present work we tested the effectiveness of two CAIs in a mouse model of paclitaxel-induced neuropathic pain. **5b** and **5d** are two selenides with a benzenesulfonamide moiety that have been shown to have the best profile in the prevention of diabetic cerebrovascular pathology [34], showing a relevant inhibition of six human (h) CA isoforms, hCA I, II, VII, IX, and particularly the mitochondrial ones VA and VB [34]. Moreover, the combination of the sulfonamide moiety with an organo-selenium scaffold is particularly interesting to bring antioxidant properties to the molecules [35,36] and it may represent a further benefit since oxidative stress and mitochondrial dysfunctions are strongly related to paclitaxel-induced neurotoxicity [37,38].

We first evaluated the potential therapeutic properties of **5b** and **5d** in a mouse model of paclitaxel-induced neuropathic pain. For this purpose, compounds were per os (p.o.) administered on day 10, when neuropathy was well established. Four paclitaxel injections significantly decreased the animal’s licking latency to 9.8 ± 0.2 s in comparison to 18.2 ± 0.4 s of the control group (vehicle + vehicle). The acute treatments with **5b** reduced the thermal allodynia in a dose-dependent manner; the dose of 100 mg/kg completely abrogated paclitaxel-induced hypersensitivity at 30 min with the anti-hyperalgesic effect that lasted up to 90 min. The dose of 30 mg/kg was still active even if the effect generated was lower with respect to that achieved with the dose of 100 mg/kg but was still statistically significant (Figure 1). **5d** showed a similar anti-allodynic profile to that evoked by **5b** with the peculiarity of the onset of the efficacy that was delayed by 15 min (Figure 1). AAZ, used as the reference drug, was administered in a single dose of 100 mg/kg. The effect exerted was lower in comparison to the other two CAIs, reaching statistical significance only 15 min after treatment (Figure 1). Of note, the toxicity of compound **5d** was tested in our previous work in which we demonstrated that RBE4 cell viability was not altered by 24-h incubation with the compound [34]. The same results were obtained with the molecule **5b** and have been reported in the Appendix A.

Thereafter, to assess the protective effect of **5b** and **5d** in the same neuropathic pain model, both compounds were subjected to a repeated treatment at 100 mg/kg over a 15-day period. Molecules were per os daily administered starting the same day of paclitaxel injection until the end of the experiment (day 15). AAZ 100 mg/kg was still used as the reference drug. The response to a thermal non-noxious stimulus (Cold plate test) was measured on days 4, 7, 11, and 15, 24 h after the last treatment. Paclitaxel injections evoked a painful condition that was maximum on day 15 in comparison to the control animals (9.0 ± 0.3 s vs. 18.6 ± 0.4 s, respectively). Both molecules were active after one week of repeated treatment, evoking an anti-allodynic effect when the Cold plate test was performed on day 7 from the beginning of the experiment. On the same day, AAZ was inactive (Figure 2). During the second week of treatment, **5b** showed the best protective profile, increasing the mouse licking latency up to 16 s on day 11. The anti-allodynic effect remained stable until the end of the experiment (day 15). **5d** exerted a lower antalgic profile in comparison to **5b** but was still significant (Figure 2).

To exclude that the sub-chronic treatment with compounds did not lead to tolerance development to the antinociceptive effect exerted in paclitaxel-treated mice, on day 15, the molecules were administered for the last time, and their acute effects were evaluated over the next two hours (Figure 3). The new daily treatments with **5b** and **5d** significantly improved the mouse pain threshold, indicating an additive anti-allodynic effect of the compounds that lasted up to 60 and 75 min, respectively (Figure 3). Conversely, AAZ was ineffective (Figure 3).

Clinically speaking, these are very significant results since other antinociceptive or analgesic drugs widely used, such as morphine, tramadol or oxycodone, present several limitations and side effects such as tolerance after repeated administration both in naïve animals [39,40,41] and in mice and rats treated with paclitaxel [42,43]. To support the active role of CAIs against chemotherapeutic side effects, in a recent work, CAIs for the IX isoform were introduced into platinum prodrugs to boost cisplatin and oxaliplatin antitumor activity, but it has also been observed that the systemic side effects of platinum drugs were consistently reduced [44].

Deepening the mechanisms that lead to paclitaxel-induced neuropathy, it is well accepted the phenomena of ganglionopathy and axonopathy in the peripheral nervous system [45] but less is known about the central mechanisms involved. Our previous work reported a complex maladaptive plasticity of astrocytes and microglia in the spinal cord as well as in the cerebral area involved in pain control [46]. These data are in line with evidence reported in other studies that highlight the pivotal role of glial cells in pain development and chronicization [47,48]. The active role of glial cells in pain development has been confirmed by treatments able to prevent glial activation that lead to pain reduction [49,50].

To evaluate the capability of **5b** and **5d** to intervene against the maladaptive plasticity that occurs in the nervous system during paclitaxel treatment, at the end of the repeated treatment with compounds (day 15), animals were sacrificed and the lumbar spinal cord was collected. In Figure 4, the immunohistochemistry of the lumbar spinal cord using the antibody against GFAP to label astrocytes is shown. The chemotherapeutic treatment significantly increased the number of GFAP-positive cells and the GFAP fluorescence intensity. **5d** and AAZ restored both of these parameters, while **5b** significantly decreased only the number of GFAP-positive cells (Figure 4).

Microglia were also altered by paclitaxel injections; all compounds tested counteracted the increase in Iba1-positive cells and of Iba1 fluorescence intensity (Figure 5). Immunohistochemical studies demonstrated that the mitochondrial isoform CA V is expressed in neurons and astrocytes but not in oligodentrocytes of several areas and tissues of the nervous system such as the sciatic nerve, spinal cord, cerebral cortex, hippocampus, and cerebellum. This distribution suggests that CA V is fundamental in physiological conditions regulating the intramitochondrial calcium level, gluconeogenesis, and neuronal transmission facilitating the bicarbonate ion-induced GABA responses in neurons [28]. The CA IV was the second isozyme found in the nervous system, and despite that its role in the nervous system is not yet understood, it is important to mention that being the CA IV mostly expressed on the capillary of endothelial cells, particularly on the luminal surface [51], it has a unique position at the blood–brain barrier that has been known to be subject to damage during chemotherapies [52].

Growing evidence over the last two decades indicates that many chemotherapeutic agents cause mitochondrial injury in the peripheral sensory nerves by disrupting the mitochondrial structure and bioenergetics, increasing oxidative stress, and altering mitochondrial transport systems. These structural and functional alterations are recognized to be at the basis of chemotherapy-induced neuropathies [22,53].

In this regard, mitochondrial dysfunction has been reported in vitro models of paclitaxel-induced damage [54]. Mitochondrial calcium accumulation, accompanied by ATP depletion and oxidative stress, can promote the opening of the mitochondrial permeability transition pore (mPTP), a multi-protein complex located between the inner membrane and outside one, responsible for the loss of ΔΨ and finally mitochondrial swelling. As a confirmation of the functional damage of peripheral nerve mitochondria in rats treated with paclitaxel, the appearance of pain was associated with a significant increase in the number of vacuolated and swollen mitochondria.

Conversely, some studies demonstrated that the majority of mitochondria in the spinal cord are not affected in paclitaxel-induced pain in rat models, but in this regard, evidence is contradictory [55].

In this context, ΔΨ is considered a reliable measurement of the mitochondrial function, and, under our experimental conditions, mitochondria isolated from cerebral tissue of animals submitted to treatment with paclitaxel showed, in agreement with the literature, a dysfunction; their ΔΨ value was significantly depolarized compared with the vehicle group (189 ± 2 mV). In particular, ΔΨ was −175 ± 1 mV, a value indicative of coupled organelles but more vulnerable against several types of injury, such as exposure to chemotherapy drugs. Mitochondria isolated from animals treated with paclitaxel and CAIs, AAZ as well as the new compounds (**5b** and **5d**), showed a ΔΨ value more negative compared with the paclitaxel group (−183 ± 3, −182 ± 2, and −180 ± 2, respectively), suggesting that exposure to CAIs may improve mitochondrial function. In this regard, **5b**, similar to AAZ, significantly improved the membrane potential. A similar profile was observed in mitochondria isolated from spinal cords of the same animals, although the data seem to be more dispersed. These results lead us to hypothesize that paclitaxel-induced mitochondrial oxidative stress may be crucial, and that the contribution of CAIs may be useful (Figure 6).

CS activity is considered a marker of cell metabolism, but being the first enzyme of the Krebs cycle, it is widely used as marker of mitochondrial function. According to the mitochondrial damage induced by treatment with paclitaxel, we observed a significant reduction in CS activity, in organelles isolated from the brain and spinal cords. On the other hand, treatment with CAIs, in particular **5b**, produced a significant improvement in CS activity, confirming the positive effects offered by this agent (Figure 7).

## 3. Materials and Methods

### 3.1. Chemistry

Compounds **5b** and **5d** were reported earlier by our group [34].

### 3.2. Cell Culture and Treatment

Rat brain endothelial cells RBE4 were obtained from the American Type Culture Collection (Rockville, MD, USA) and were cultured in MEM Alpha/NutriHam F-10 in 1:1 ratio (Thermo Fisher Scientific, Milan, Italy), supplemented with 10% fetal bovine serum, 0.1% basic fibroblast growth factor, 100 IU mL^−1^ penicillin, and 100 μg mL^−1^ streptomycin (Sigma, Milan, Italy) at 37 °C in a humidified, 5% CO_2_ atmosphere; 4 × 10^4^ cells per well were plated in 96-well plates and treated for cytotoxicity assay (cells were incubated with the tested compound for 24 h).

### 3.3. Cell Viability Assay

Cell viability was assessed using 3-(4,5-dimethylthiazol-2-yl)-2,5-diphenyltetrazolium bromide (MTT) assay. Following treatments, cells were washed and incubated with MTT solution (1 mg/mL) at 37 °C for 30 min in a humidified, 5% CO_2_ atmosphere. After washing, the formazan crystals were solubilized in 200 μL DMSO, and absorbance was measured at 550 nm.

### 3.4. Animals

CD-1 mice (Envigo, Varese, Italy) weighing approximately 20–25 g at the beginning of the experimental procedure were used. Animals were housed in the Centro Stabulazione Animali da Laboratorio (Ce.SAL; University of Florence, Florence, Italy) and used at least one week after their arrival. Ten mice were housed per cage (size 26 cm × 41 cm); animals were fed with a standard laboratory diet and tap water ad libitum, kept at 23 ± 1 °C, with a 12-h light/dark cycle (light at 7 A.M.).

### 3.5. Paclitaxel Mouse Model of Neuropathy

Two mg/kg paclitaxel (Carbosynth, Pangbourne, UK) was dissolved in a mixture of 10% saline solution and Chremophor EL, a derivative of castor oil and ethylene oxide that is clinically used as a paclitaxel vehicle. The drug was injected intraperitoneally (i.p.) on days 1, 3, 5, and 8 [56,57]. Control animals received an equivalent volume of the vehicle.

### 3.6. Treatments

When neuropathy was fully established (day 10), **5b** and **5d** were suspended in 1% solution of carboxymethylcellulose sodium salt (CMC) and per os (p.o.) acutely administered at doses of 30 and 100 mg kg^−1^ to evaluate their symptomatic effect. The synthesis of the compounds was previously reported in [34]. The measurement of thermal allodynia was performed before and 15, 30, 45, 60, 75, 90, 105, and 120 min after treatments. Acetazolamide (AAZ; 100 mg/kg, p.o.) was used as the CAI reference drug. Afterwards, to highlight a protective effect, repeated per os administrations of 100 mg/kg of **5b** and **5d** were carried out daily from the beginning of paclitaxel administration (day 1) to the end of the experiment (day 15). The measurement of thermal allodynia was performed on days 4, 7, 11, and 15 from the beginning of the experiments, 24 h after daily treatments. Moreover, on day 15, the Cold plate test was also performed after the new daily treatment with compounds at 30, 45, 60, 75, and 90 min. Control animals were treated with the vehicle (CMC 1%).

### 3.7. Cold Plate

Thermal allodynia was assessed using the Cold plate test. With minimal animal–handler interaction, mice were taken from home-cages and placed onto the surface of the Cold-plate (Ugo Basile, Varese, Italy) maintained at a constant temperature of 4 °C ± 1 °C. Ambulation was restricted by a cylindrical Plexiglas chamber (diameter: 10 cm, height: 15 cm) with an open top. A timer controlled by a foot pedal began timing response latency from the moment the mouse was placed on the cold surface. Pain-related behavior (licking of the hind paw) was observed, and the time (seconds) of the first sign was recorded. The cut-off time of the latency of paw lifting or licking was set at 30 s [58,59].

### 3.8. Immunohistochemistry

On day 15 of treatment performed to evaluate the protective effect of compounds, after the behavioral pain measurements, mice were sacrificed and the lumbar spinal cord segments were removed, post-fixed in 4% paraformaldehyde, and then cryoprotected in 30% sucrose solution at 4 °C. Slide-mounted cryostat sections (5 µm) were processed for indirect immunofluorescence histochemistry. Formalin-fixed cryostat sections (5 µm) were incubated for 1 h in blocking solution (Bio-Optica, Milan, Italy) at room temperature and were then incubated for 24 h at 4 °C in PBST containing rabbit primary antisera diluted 1:1000 and 5% normal donkey serum. The primary antibody was directed against Iba1 (rabbit, 1:1000; Wako Chemicals, Richmond, VA, USA) for microglial staining and against glial fibrillary acidic protein (GFAP; mouse, 1:5000; Chemicon, Temecula, CA, USA) for astrocyte staining. After rinsing in PBST, sections were incubated in donkey anti-rabbit IgG secondary antibody labelled with Alexa Fluor 568 (1:1000, Invitrogen, Carlsbad, CA, USA) for microglia and Alexa Fluor 488 (1:500, Invitrogen, Carlsbad, CA, USA) for astrocytes at room temperature for 1 h. Negative control sections (no exposure to the primary antisera) were processed concurrently with the other sections for all immunohistochemical studies. We obtained a single optical density value for the dorsal horns by averaging the two sides in each rat, and these values were compared to the homologous average values from the vehicle-treated animals. Images were acquired by a motorized Leica DM6000B microscope equipped with a DFC350FX camera (Leica, Mannheim, Germany). Quantitative analysis of GFAP and Iba1-positive cells was performed by collecting at least three independent fields through a 20× 0.5NA objective. GFAP-positive cells were counted using the “cell counter” plugin of ImageJ, while Iba1-positive cells were quantified by means of the automatic thresholding and segmentation features of ImageJ. The GFAP signal in immunostained sections was quantified using FIJI software (distributed by ImageJ, NIH, Bethesda, MD, USA) by automatic thresholding images with the aid of the “Moments” algorithm, which we found to provide the most consistent pattern recognition across all acquired images. Results (not shown), given as the area fraction (%) occupied by the thresholded GFAP signal, revealed a common trend between GFAP expression and astrocyte cell number. Five spinal cord sections were analyzed for each animal.

### 3.9. Statistical Analysis

Behavioral measurements were performed on ten mice for each treatment carried out in two different experimental sets. All assessments were made by researchers blinded to animal treatments. Results are expressed as the mean ± (S.E.M.) with one-way analysis of variance. A Bonferroni’s significant difference procedure was used as a post hoc comparison; *p*-values < 0.05 or <0.01 were considered significant. Data were analyzed using Origin 9 software (OriginLab, Northampton, MA, USA).

### 3.10. Citrate Synthase (CS) Activity

Brain and spinal cord fragments from animals treated with paclitaxel, in the presence or absence of CAIs, were homogenized in isolation buffer (composition: sucrose 250 mM, Tris 5 mM, EGTA 1 mM, Triton X-100 0,02%; pH 7.4) using a GentleMACS dissociator (Miltenyi Biotec, Bologna, Italy). The homogenates obtained were centrifuged at 12,000× *g* for 15 min at 4 °C (Sigma 3-18KS, Osterode am Harz, Germany). The supernatant was used to measure the activity of CS. The protein concentration in the supernatant was determined spectrophotometrically (EnSpire, PerkinElmer, Waltham, MA, USA) by the Bradford assay. The enzymatic reaction was performed in 10 μg/mL of protein (Trizma base 100 mM, 5,5′-dithiobis-(2-nitrobenzoic) acid 100 μM, acetylcoenzyme A 100 μM, and oxaloacetic acid 500 μM) as previously described [60]. The kinetics were evaluated spectrophotometrically at 412 nm every 30 s for 15 min. The isolated enzyme (Sigma–Aldrich, St. Louis, MO, USA) was used for the calibration line.

Data analysis. CS activity was evaluated on five brain and five spinal cord tissue samples per group. Enzymatic activity was expressed in mU/mL. Data analysis was performed using GraphPad Prism 7.0 software. Student’s *t*-test was used for the statistical analysis (*p* < 0.05 was considered as the limit of statistical significance).

### 3.11. Mitochondrial Membrane Potential

Brain tissue or spinal cord fragments, from animals treated with paclitaxel, in the presence or absence of CAIs, were placed in MSE (composition: mannitol 225 mM, sucrose 75 mM, HEPES 5 mM, EGTA 1 mM, BSA 1 mg/mL) and finely cut. The tissue was manually homogenized in 5 mL of MSE, preventing air formation, and centrifuged at 1000× *g* for 3 min at 4 °C. The supernatant was stored on ice. The pellet was resuspended in 5 mL of MSE and centrifuged again under the same conditions. Each supernatant was centrifuged at 10,000× *g* for 10 min at 4 °C. The pellets were recovered and resuspended in 5 mL of MS (composition: MSE without EGTA) plus 0.01% *w*/*v* digitonin. The suspension obtained was centrifuged again at 10,000× *g* for 10 min at 4 °C. The pellet was resuspended in 400 µL of MS. Bradford assay was used to determine the protein concentration in the suspension. The analysis of the mitochondrial membrane potential (ΔΨ) was carried out on the obtained mitochondrial suspension. Mitochondria were incubated with rhodamine (5 nM) in 96-well black plates. This test was performed in triplicate at a mitochondrial concentration of 50 µg/mL. The analysis was performed under fluorescence and followed every 30 s for 5 min (10 total readings). The ΔΨ was calculated with the Nernst equation:$$\Delta \Psi =60\times log\frac{{\left[X\right]}_{in}}{{\left[X\right]}_{out}}$$

Data analysis. ΔΨ was evaluated on five brain and five spinal cord tissue samples per group. GraphPad Prism 7.0 software was used for the analysis. Student’s *t*-test was used for the statistical analysis (*p* < 0.05 was considered as the limit of statistical significance).

## 4. Conclusions

In conclusion, we reported the effectiveness of two selenides bearing benzenesulfonamide moieties, named **5b** and **5d**, against a paclitaxel mouse model of neuropathy. In particular, the therapeutic and protective properties were highlighted after acute and sub-chronic treatments, respectively. Moreover, **5b** and **5d** were able to counteract the maladaptive plasticity of glial cells and to reduce the mitochondrial dysfunction affecting the central nervous system. These results support the use of CAIs as a therapeutic option for the management of neuropathic pain; in particular, these data lay the foundation for deepening the role of the isoforms VA and VB in the development of chemotherapy-induced neuropathic pain. Future studies should therefore be carried out with even more selective molecules in inhibiting the CA mitochondrial isoforms to understand the real therapeutic potential against neuropathic pain.

## Data Availability

The data presented in this study are available on request from the corresponding author.

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
