# Peer review of "Inhibitors of Mitochondrial Human Carbonic Anhydrases VA and VB as a Therapeutic Strategy against Paclitaxel-Induced Neuropathic Pain in Mice"

_ijms, 2022, doi:10.3390/ijms23116229_

Round 1
Reviewer 1 Report
Laura et al reported that mitochondrial human carbonic anhydrases VA and VB possessed therapeutic and protective effects against paclitaxel-induced neuropathy without tolerance development. This study is interesting, data are sound. However, there are several issues that need to be addressed. A revision is suggested.
- The toxicity of 5b and 5d needs to be investigated and presented in this study.
- Please also discuss the dosage used in this study and discuss the physiological concentration of 5b and 5d.
- Both in vivo and in vitro assays were presented, however, the mechanism is still largely unclear. The specific inhibitor or siRNA should be used to study the mechanism.
- Please also study mitochondrial biogenesis.
- Please discuss the limitation of this study.
- Please use JC1 for the Mitochondrial membrane potential assay.
Author Response
Reviewer 1
Reviewer #1: Micheli et al reported that mitochondrial human carbonic anhydrases VA and VB possessed therapeutic and protective effects against paclitaxel-induced neuropathy without tolerance development. This study is interesting, data are sound. However, there are several issues that need to be addressed. A revision is suggested
Point 1. The toxicity of 5b and 5d needs to be investigated and presented in the study.
Response: We are grateful to the reviewer for the revision of our manuscript and we perfectly agree with the request regarding the toxicity of 5b and 5d. As mentioned in the main manuscript, the design and synthesis as well as the inhibitory action against six human carbonic anhydrase isoforms of these molecules have been already investigated and reported in a previous work (Angeli et al., ACS Med Chem Lett. 9(5): 462–467. doi: 10.1021/acsmedchemlett.8b00076). In this paper, the toxicity of compound 5d has been also investigated. In particular the RBE4 cell viability following 24 h incubation with compound 5d at different concentrations (10 mM, 30 mM and 100 mM) were tested. In this assay,5d resulted safe at all dosages. We reproduced the same test for compound 5b that resulted safe as well as compound 5d, the data obtained have been reported in the table below and were also added in the main text in the result section.
|
|
Cell viability % |
|
5d concentration (mM) |
24 h incubation |
|
0 |
100 ± 2.3 |
|
10 |
98.4 ± 1.6 |
|
30 |
102.6 ± 1.8 |
|
100 |
94.6 ± 5.3 |
RBE4 cells (4 × 104 cell/well) were treated with increasing concentrations of 5b (10 μM, 30 μM and 100 μM). Incubation was allowed for 24 h. Cell viability was measured by the MTT assay. The control condition was arbitrarily set as 100 % and values expressed as mean ± SEM of three experiments.
Point 2. Please also discuss the dosage used in this study and discuss the physiological concentration of 5b and 5d.
Response. We thanks the referee for the possibility to clarify this point. For the dosages used in the this study, and particularly in the first stage in which we evaluate the therapeutic effect of the compounds (acute treatments), we based on the results obtained in our previous works in which we highlighted the pain relieving properties of similar carbonic anhydrases inhibitors in the animal model of neuropathic pain induced by the chemotherapeutic drug oxaliplatin (Tanini, et al. Eur. J. Med. Chem. 2021, 225, 113793, doi:10.1016/j.ejmech.2021.113793; Nocentini et al. J. Med. Chem. 2020, 63, 5185–5200, doi:10.1021/acs.jmedchem.9b02135; Akgul et al., Eur. J. Med. Chem. 2022, 227, 113956, doi:10.1016/j.ejmech.2021.113956. In these cited articles the doses used were 10 mg/kg, 30 mg/kg and 100 mg/kg that showed a pain relieving effect in a dose-dependent manner. Regarding the physiological concentration of 5b and 5d,we don’t have already this information since this is wants to be a proof of concept study. Surely plasma and CNS concentrations of compounds will be performed on further pharmacological studies in which we will deeply investigate the effect of these compounds against paclitaxel-induced neuropathy.
Point 3. Both in vivo and in vitro assays were presented, however, the mechanism is still largely unclear. The specific inhibitor or siRNA should be used to study the mechanism.
Response. We perfectly agree with the reviewer and with the possibility to deeper investigate the mechanism of action of the compounds 5b and 5d. Unfortunately we are not able to use specific antagonists or siRNAs to better clarify the mechanism since the molecules used in the study are inhibitors of carbonic anhydrase isoforms and not agonists. As inhibitors we can not block the site of action where the compounds act in order to confirm their pharmacodynamics mechanism.
Point 4. Please also study mitochondrial biogenesis.
Response: Thanks for your comment, in this study we analysed the activity of citrate synthase as a measure of the mitochondrial metabolism, a parameter closely related to the mitochondrial function. However, a deeper investigation of mitochondrial biogenesis, throgouth the evaluation of the expression of the genes involved in its function deserves to be carried out, certainly, in the future we will further explore this aspect.
Point 5. Please discuss the limitation of this study.
Response. According to the reviewer, the limit of this study has been added in the conclusion section.
Point 6. Please use JC1 for the Mitochondrial membrane potential assay.
Response: we thanks for the suggestion. JC1 is a well-known very sensitive probe, that exhibits potential-dependent accumulation in mitochondria and changes its colour on the basis of monomeric or dimeric organization. Indeed, when mitochondrial membrane potential is very negative (about -180 mV) an high amount of JC1 will be accumulated into the matrix, producing a red fluorescence (as result of j-aggregated form). Instead, a mitochondrial membrane potential depolarized will promote a reduced uptake of JC1 into the matrix, that will remain outside the mitochondria and it will be emissed a green fluorescence (monomeric form). In order to appreciate this behaviour and to calculate the value of mitochondrial membrane potential a cytofluorimetric measurement should be necessary. Unfortunately, we are not in condition to use this type of approach. However, a great body of evidence suggests that also other types of probes may successful be used for this type of acquisition, and rhodamine is one of the probe used since more time. Moreover, we performed this approach on isolated mitochondria from tissues, then non-specific interactions may be excluded.

Reviewer 2 Report
Carbonic anhydrases (CAs and EC 4.2.1.1) are a family of ubiquitous Zn2+ containing enzymes which catalyze the conversion of CO2 and H2O to HCO3- with the release of a proton. They have vital roles in pH regulation and in the balance of body fluids and are involved in homeostasis regulation and cellular respiration. Dysregulation of CAs could lead to many diseases including cancer. CA inhibitors are used in clinical practice, as diuretics and for treating glaucoma. Furthermore, they are being investigated for application such as obesity, cancer, and epilepsy.
In neuropathic pain conditions, the mitochondrial CA VA and CA VB isoforms are of interest as they regulate the respiration rate and reactive oxygen species (ROS) production.
In this manuscript the authors evaluated the efficacy of compound 5b, [4-(phenethylselanyl)benzenesulfonamide] and compound 5d, [4-(isopropylselanyl)benzenesulfonamide], which are two selenides bearing the benzenesulfonamide moiety, against the CA VA and VB isoforms, in a mouse model of paclitaxel-induced neuropathic pain.
In general, the manuscript is well written and presented. The experiments are well conducted, the results are interesting and have translational potential.
Comments:
- Please provide the source for compounds 5b and 5d.
- Since 5b and 5d contain the sulfonamide moiety, I wonder if these compounds release hydrogen sulfide. If they do, this would explain a lot of the observations reported here. It should not be a problem or take long to check this in vitro.
Author Response
Reviewer 2
Reviewer #2: Carbonic anhydrases (CAs and EC 4.2.1.1) are a family of ubiquitous Zn2+ containing enzymes which catalyze the conversion of CO2 and H2O to HCO3- with the release of a proton. They have vital roles in pH regulation and in the balance of body fluids and are involved in homeostasis regulation and cellular respiration. Dysregulation of CAs could lead to many diseases including cancer. CA inhibitors are used in clinical practice, as diuretics and for treating glaucoma. Furthermore, they are being investigated for application such as obesity, cancer, and epilepsy.
In neuropathic pain conditions, the mitochondrial CA VA and CA VB isoforms are of interest as they regulate the respiration rate and reactive oxygen species (ROS) production.
In this manuscript the authors evaluated the efficacy of compound 5b, [4-(phenethylselanyl)benzenesulfonamide] and compound 5d, [4-(isopropylselanyl)benzenesulfonamide], which are two selenides bearing the benzenesulfonamide moiety, against the CA VA and VB isoforms, in a mouse model of paclitaxel-induced neuropathic pain.
In general, the manuscript is well written and presented. The experiments are well conducted, the results are interesting and have translational potential.
Comments
Point 1. Please provide the source for compounds 5b and 5d.
Response. We are grateful to the reviewer for the revision of our manuscript and for the positive feedbacks received. We perfectly agree with the reviewer, the source of compound has been added in Material and Methods section.
Point 2. Since 5b and 5d contain the sulfonamide moiety, I wonder if these compounds release hydrogen sulfide. If they do, this would explain a lot of the observations reported here. It should not be a problem or take long to check this in vitro.
Response. The sulfonamide moiety is well-studied and explored group in the field of medicinal chemistry, to date, no one has reported its possible release of H2S in physiological environments. Chemically, the sulfonamide group is very stable. In order to change its oxidation state +6 to -2, a strong reducing agent is required and in large quantities, which we do not find in physiological environments. Even if we did, to obtain the SH2 group an aromatic nucleophilic substitution would have to take place, which is highly improbable under physiological conditions.

Round 2
Reviewer 1 Report
Accept in present form